# Proposal and Validation of a New Nonradiological Method for Postoperative Three-Dimensional Implant Position Analysis Based on the Dynamic Navigation System: An In Vitro Study

**DOI:** 10.3390/jpm13020362

**Published:** 2023-02-18

**Authors:** Feifei Ma, Mingyue Liu, Xiaoqiang Liu, Tai Wei, Lilan Liu, Feng Sun

**Affiliations:** 1First Clinical Division, Peking University School and Hospital of Stomatology, Beijing 100034, China; 2Department of Prosthodontics, Peking University School and Hospital of Stomatology, Beijing 100081, China

**Keywords:** dental implants, dynamic navigation, postoperative implant position evaluation, computer-guided implantology

## Abstract

Purpose: To propose a novel, radiation-free method for postoperative three-dimensional (3D) position analysis of dental implants based on the dynamic navigation system (DNS) and evaluate its accuracy in vitro. Methods: A total of 60 implants were digitally planned and then placed in the standardized plastic models with a single-tooth gap and a free-end gap under the guidance of the DNS. Postoperative 3D positions of the inserted implants were evaluated using specially designed navigation-based software, and its datasets were superimposed onto those of cone beam computed tomography (CBCT) for accuracy analyses. Deviations at the coronal, apical, and angular levels were measured and statistically analyzed. Results: The mean 3D deviation was 0.88 ± 0.37 mm at the entry point and 1.02 ± 0.35 mm at the apex point. The mean angular deviation was 1.83 ± 0.79 degrees. No significant differences were noted in the deviations between implants placed in the single-tooth gap and the free-end situation (*p* > 0.05) or between different tooth positions at distal extensions (*p* > 0.05). Conclusions: This non-radiographic method provides facile, efficient, and reliable postoperative implant position evaluation and may be a potential substitute for CBCT, particularly for implants placed under the guidance of dynamic navigation.

## 1. Introduction

As part of a coordinated surgical and prosthetic concept, the high accuracy of implant placement plays an increasingly important role towards reducing biological complications and improving the aesthetic and functional performance of restorations [1,2]. After implant surgery, it is necessary to identify the position of the implant and its relationship with the surrounding anatomical structures (inferior alveolar nerve, mental foramina, etc.) through radiological examination. In two-dimensional (2D) radiographic imaging, analyzing images with blurred or overlapping features is challenging, making accurate evaluations difficult [3]. In contrast, cone beam computed tomography (CBCT) images provide distortion-free and metrically accurate spatial diagnostics and have become the dominant method for identifying postoperative implant positions in three dimensions (3D) [4]. However, postoperative CBCT induces an additional radiation exposure of 19–368 mSv [5] and incurs higher costs. Moreover, the metal artifacts caused by implants on CBCT may affect assessment accuracy [6].

To overcome the shortcomings of radiographic analysis, alternative methods have been established. In a study by Nickenig and Eitner [7], an indirect method based on CBCT images of master casts with implant analogs was used; however, the results showed great variance in precision. Additionally, this method requires complicated procedures. Thereafter, with the development and wide application of digital technology, the use of intraoral scanning to evaluate the 3D implant position after surgery has been extensively reported [3,4,8,9,10,11]. In this method, a scan body is installed at the implant site after surgery, an intraoral digital impression is acquired, and the position of the inserted implant is then calculated using computer-guided implant software [3]. Previous investigations have revealed that this method can help assess the implant location with acceptable accuracy [3,4,8,9,10,11]. Nevertheless, the conclusion was mainly based on studies with small sample sizes [3,8,9]. Moreover, additional equipment and time are required for postoperative digital impression and data superimposition. In addition, it is unsuitable in certain situations, such as in patients with edentulism.

In recent years, computer-guided implant dentistry has developed rapidly, with the potential for more predictive and less invasive implant placement [12,13,14,15]. Dynamic navigation is one of the most recent innovations in this field. The dynamic navigation system (DNS) can improve the precision of implant placement through accurate preoperative design and precise intraoperative surgery guidance [16]. When CBCT scans are incorporated into the DNS, the bone structures and optimized reconstructions can be displayed by the computer-assisted design software during the preoperative phase, thus improving treatment planning. More importantly, dynamic navigation allows for the intraoperative monitoring of implant placement in real time using specific software. The accuracy of implant placement under the guidance of the DNS has been assessed in substantial studies, and the results have shown that it is more accurate than the traditional freehand method and comparable to template-based static guidance [17,18,19,20]. Although additional equipment cost is required, the DNS possesses several advantages compared to static navigation [21], including the omission of fabricating static guides, higher visualization, flexibility, cooling capacity during drilling, and the possibility of application in cases of reduced mouth opening. Moreover, it may minimize morbidity rates by increasing the likelihood of flapless surgery. Owing to these benefits, the use of the DNS in dental practice has rapidly increased. Stopp et al. [22] described a non-radiographic method for postoperative assessment of the 3D position of an implant inserted under the guidance of the DNS; however, this method requires a specially fabricated device and is not sufficiently precise to conclusively determine the implant position.

The purpose of the present study was to present a novel method for identifying the postoperative implant position in 3D using the DNS and to evaluate the accuracy of this DNS-based method in vitro. This method can help generate the 3D geometry of the implant fixture immediately after navigation-guided surgery with specially designed software integrated in the DNS, thus helping to avoid additional postoperative radiographic evaluations. The secondary aim was to assess the difference between implants placed in a single-tooth gap and a free-end gap, with the null hypothesis stating that there would be no difference in accuracy between the two situations.

## 2. Materials and Methods

### 2.1. Study Model

This in vitro research was conducted on 20 standardized plastic dentition models (Dongguan Lixiang Model Science Inc., Dongguan, China; Figure 1A) representing partially edentulous mandibular jaws (Kennedy Class II) with a free-end gap at the left mandible corresponding to FDI teeth nos. 35–37 and a single-tooth gap at the right mandible corresponding to FDI tooth no. 46. To simulate the clinical scenario, the plastic models were reproducibly fixed onto dental simulation units (NL-2000; Nissin Inc., Kyoto, Japan; Figure 1B) with an elastic gingival mask before implant placement.

The sample size was calculated by using the Clin Epi software (Peking University Third Hospital, Beijing, China). Considering a power of 0.90, an alpha value of 0.05, and the results obtained from the pre-experiment (the angular deviation of STG was 1.69 ± 0.59°, and that of DE was 2.21 ± 0.61°), a minimum sample size of 20 per group was calculated, and, therefore, a total of 60 implants were required.

### 2.2. Digital Planning of Implant Position

Preoperative CBCT images were acquired under uniform conditions (Carestream 9300; Carestream Health, Toronto, ON, Canada; 75 kV, 4 mA). The tomographs were exported in a Digital Image for Communication in Medicine (DICOM) file format with a slice thickness of 0.8 mm. The DICOM files were subsequently imported into the dynamic navigation software (Dcarer^®^, DHC-D12, Suzhou Digital-health care Co., Ltd., Suzhou, China), and the implant positions were determined according to the desired reconstruction using the planning module (Figure 2). Three implants were planned for each model to achieve the calculated sample size of sixty.

### 2.3. Experimental Implant Placement

Before implant placement, the infrared tracking markers of the DNS were fixed on the plastic models using an appropriate splint and self-curing resin to receive infrared light from the stereo camera of the DNS. The handpiece and handpiece tracker were assembled and registered using the tracking points. The infrared tracking plate was then calibrated, and the plastic model was subsequently matched to the preoperative virtual design using CBCT. The cusp calibration method, which was used in the present study [23], is shown in Figure 3. This method involved using the cusps or fossae of the remaining adjacent teeth as markers. Specifically, the cusps of the right mandibular second molar, first and second premolars, and left mandibular first premolar were used as markers. The spatial positions of the handpiece and plastic model were continuously tracked by the stereo camera (Figure 1B), and the DNS then calculated the positions of the handpiece and the model relative to the presurgical plan.

Sequential drilling for osteotomies was performed based on the recommended drilling protocol. After all the implant cavities were prepared, three implants (bone level implant: SLA; length: 10 mm; diameter: 4.1 mm; Straumann GmbH, Basel, Switzerland) were subsequently inserted on each model at a torque of 25 Ncm. The DNS was used throughout the entire length of the implantation process to guide the surgeon in real time by optically tracking the handpiece position with the stereo camera and providing visual feedback on the screen (Figure 4). All implants were placed by a single surgeon with more than ten years of clinical experience in implant surgery and more than three years of experience using the DNS. The number of implants placed by the surgeon with the DNS was about 500 in vivo and 150 in vitro.

### 2.4. Accuracy Evaluation

The data for the accuracy evaluation of each achieved implant were acquired using the DNS-based method and postoperative CBCT.

Once the implant was in place, the DNS detected the location of the handpiece and, therefore, that of the implant carrier (Figure 5A). Accordingly, the navigator automatically calculated the position of the implant using specially designed software (Figure 5B; Postoperative CBCT Simulation System; Suzhou Digital-health care Co., Ltd., Suzhou, China) and projected a virtual implant into the dataset to correspond with the 3D position of the implant carrier. The reconstructed 3D position image of the virtual implant was fused with the preoperative CBCT image automatically, and, ultimately, a 3D image representing the position of the inserted implant was generated (Figure 6).

Postoperatively, a radiological assessment of the inserted implants was performed using CBCT. The settings were consistent with those of the preoperative scans. The resulting volume datasets were imported into the dynamic navigation accuracy verification software as DICOM files. The reconstructed 3D images containing visible representations of the achieved implants generated by the DNS-based method were superimposed on the postoperative CBCT dataset using clearly defined anatomical landmarks (the cusps of the right mandibular second molar, first and second premolars, and left mandibular first premolar) that could be located in both sets of data. The superimposition was achieved through a mixture of manual point selection and automatic alignment by the software. The software quantified the accuracy and measured the deviations between the two methods based on the alignment (Figure 7). Target parameters included 3D deviation at the entry and apex points, 2D horizontal and vertical deviation at the entry and apex points, and angular deviation (Figure 8). CBCT served as a reference for determining deviation.

### 2.5. Statistical Analysis

A descriptive analysis was conducted to evaluate the differences between the implant position in the CBCT image and the position obtained using the DNS-based method. The data are presented as mean and standard deviation (SD) values. A 95% confidence interval was used for each parameter. The Shapiro–Wilk test was performed to assess the normality of the distribution of the data. When the normality assumption was satisfied, the independent sample *t*-test was used to identify significant differences between the two sets of observations (tooth no. 35 vs. tooth no. 36). When the data were not normally distributed (deviations in the single-tooth gap and free-end gap groups), a statistical evaluation was performed using the nonparametric Mann–Whitney U test. Statistical analyses were performed using SPSS 23.0 (IBM Corp., Armonk, NY, USA). The level of significance was set at *p* < 0.05.

## 3. Results

A total of 60 implants were inserted in a unilateral free-end gap situation of the mandibular jaw (Kennedy Class II) consisting of 20 intercalated (tooth no. 46) and 40 free-end implants (tooth no. 35 and tooth no. 36).

The definitive implant position set using the DNS-based method was assessed for accuracy. The measurements of deviation between the DNS-based evaluation method and postoperative CBCT for the achieved implant position are summarized in Table 1. The 3D deviation was 0.88 ± 0.37 mm at the entry point and 1.02 ± 0.35 mm at the apex point. The angular deviation was 1.83 ± 0.79°. The 2D horizontal deviation was 0.59 ± 0.26 mm at the implant shoulder and 0.94 ± 0.42 mm at the apex. The 2D vertical deviation was 0.17 ± 1.10 mm at the implant shoulder and 0.32 ± 1.31 mm at the apex.

Table 2 shows the horizontal deviation directions of implant positions obtained using the DNS-based method. The position of the implant derived from the DNS-based method showed a comparable entry deviation in the buccal-mesial, lingual-mesial, and lingual-distal directions compared to the position obtained by CBCT. The lowest number of implants was deviated in the buccal-distal direction at the entry point. In terms of apex deviation, lingual deviation (65%) was more prominently displayed than buccal deviation.

The differences between implant position deviations in single-tooth gaps (STG) and distal extension situations (DE) are summarized in Table 3. The mean vertical deviation at the apex point was higher in the DE group than in the STG group; however, the difference was not significant (*p* = 0.122). The mean values of the other parameters also showed no statistically significant differences (*p* > 0.05). Considering different tooth positions at the free-end gap, no significant differences were noted in any measured parameters between the regions of tooth no. 35 and tooth no. 36 in terms of the deviations of postoperative implant position analyzed depending on the navigation calibration methods and CBCT (Table 4; *p* > 0.05).

## 4. Discussion

Proper positioning of an implant is crucial for short- and long-term success [1,2]. Evaluating the orientation of the implant and the distance between the implant fixture and the surrounding anatomical structures in 3D after implant installation is important. Notably, with the increased application of digital 3D diagnostic and therapeutic modalities, the analysis of the implant position achieved by static or dynamic navigation is logical. This is particularly relevant for the pre- and postoperative comparison and evaluation of the surgeon’s manual performance under digital guidance. In scientific studies, CBCT has been the most frequently used method for assessing implant positions after surgery [4]. However, radiation exposure and the additional economic burden limit its clinical application. To address these concerns, this article presents an alternative approach for 3D implant position assessment after surgery using the DNS. To the best of our knowledge, the application of the DNS to generate postoperative implant positions has not yet been reported.

Previous studies have used the intraoral scanning-assisted method, which has been proposed along with the use of static implant guides, to identify the positions of the placed implants. Several studies have investigated the reliability of the intraoral scanning method, and its accuracy has been shown to be comparable to that of CBCT [3,4,8,9,10]. However, in most reported studies, the evaluation was conducted using several animals [8,9] or a single clinical case [3]. In two previous studies by Skjerven et al. [4] and Franchina et al. [10], the accuracy was evaluated by calculating the differences between the data acquired by intraoral scanning and postoperative CBCT. These were indirect comparisons based on deviations between the planned and realized implant positions, and the increased steps of alignment and potential errors derived from direction deviations may lead to decreased accuracy. Moreover, the intraoral-based method requires additional intraoral scanning after surgery with the scan bodies correctly installed, and the manual alignment of the scan surface with the preoperative CBCT is time-consuming. Additionally, because the fusion of the scan surface with the CBCT dataset requires the use of natural teeth as landmarks, this method is inapplicable for edentulous jaws [8].

In recent years, the DNS has been increasingly used in implant surgery owing to its benefits, such as allowing for precise implant placement without the use of static guides [16]. The present study proposes a radiation-free method for postoperative 3D position evaluation of navigated inserted implants. Furthermore, its accuracy was evaluated by measuring the deviations between 3D datasets obtained using the DNS-based method and postoperative CBCT. The use of postoperative CBCT images as a reference to calculate the deviations directly may lead to a more accurate evaluation of the postoperative implant position analysis method with reduced superimposition times compared with the indirect method mentioned above. The results showed that the average 3D deviation was 0.95 mm (0.88 ± 0.37 mm at the implant shoulder and 1.02 ± 0.35 mm at the apex), and the deviation for implant inclination was 1.83 ± 0.79 degrees. The 2D deviation findings revealed that the horizontal and vertical deviations at the entry point were 0.59 ± 0.26 mm and 0.17 ± 1.10 mm, respectively. At the apical point, the values evaluated were 0.94 ± 0.42 mm and 0.32 ± 1.31 mm, respectively. From a clinical point of view, the apical deviation of the installed implant is particularly important for the prevention of severe damage to vital anatomy [24]. Postoperative implant position assessment should consider inaccuracies and identify the minimum distance between the achieved implant apex and anatomical structures as a safety measure. In the present study, the linear vertical deviation at the apex was 0.32 mm, with a confidence interval of −0.02–0.65 mm. According to these results, the most apical part of the achieved implants obtained using the DNS-based method should maintain a safe distance identical to the upper limit of the confidence interval, which was 0.65 mm in the vertical direction.

The trueness of the DNS-based method applied to free-end situations and interdental gaps was also compared in the present study. The results supported the null hypothesis, that is, no significant differences were noted in the 3D or 2D deviations between the implants placed in a single-tooth gap or a free-end gap. Regarding the most important apical deviations, implants placed in the free-end gap revealed higher linear vertical deviations than those in the single-tooth gap; however, these differences were not statistically significant. Similarly, there were no significant differences between different tooth positions (tooth no. 35 and tooth no. 36) at distal extensions, despite theoretical expectations of errors derived from the calibration process when considering the positional distance from the reference point [24]. According to these results, the accuracy of the DNS-based postoperative implant analysis method was not influenced by the position or distribution of the inserted implants.

The error of the DNS-assisted method mainly comes from the intrinsic errors of the DNS. The accuracy of the DNS in guiding implant surgery has been assessed in extensive scientific studies [13,20,25,26]. The results demonstrated that it was more accurate than free-hand implantation: the mean differences between the actual and planned implants were reported as 0.71–1.01 mm at the implant shoulder and 1.00–1.83 mm at the apex, with a deviation of 2.26–5.59 degrees [13,15,20,27]. However, the deviation measured in the DNS-guided implant surgery is the combined result of all errors in treatment planning and implant placement, including data acquisition, data processing, registration, and calibration [28]. When the DNS is used to identify the implant’s position after surgery, the possible sources of error vary slightly. The primary error source is the intrinsic errors of the DNS. Some other factors may also contribute to its inaccuracy, such as the algorithm of the specially designed software and the connection stability between the implant carrier and the implant. In the present study, the mean deviations of the 2D vertical deviations were relatively low, whereas the range of deviations was high, resulting in high maximum deviations (2.47 mm at the entry depth and 3.43 mm at the apex depth). Nevertheless, their 95% confidence intervals were −0.11–0.45 mm and −0.02–0.65 mm, respectively. However, the standard deviations of the other 2D and 3D parameters were relatively normal. This can only be attributed to the intrinsic errors of the DNS and specially designed software and needs to be improved with the optimization of the system. Additionally, the accuracy of the DNS-based evaluation method is unaffected by the movement or mouth opening extent of the patient. Based on this perspective, an in vitro study can reflect the accuracy of the method properly.

CBCT is the most popular choice for postoperative implant position analysis because it allows for a sufficiently precise representation of the hard tissues. However, noise distortion and metallic artifacts occur around dental implants, which may cause an average measurement error of 0.49 mm [6]. In the present study, it cannot be entirely excluded that the deviations between the implant position calculated by the DNS-based method and the radiologically detected position are due to the inaccuracy of CBCT scans. Further research is needed to explore this potential source of inaccuracy.

The present DNS-assisted implant position analysis method is simple and efficient, with the ability to generate 3D implant imposition images immediately after implant insertion without additional digital scanning and manual data fusion procedures, such as with the intraoral scanning-based method. Moreover, this method is effective in both dentate and edentulous patients. In addition, it could be used to determine the difference between the planned and realized position of the implant. Thus, the DNS can assist surgeons in the entire implantation process, including preoperative design, intraoperative surgery guidance, and postoperative assessment. Nevertheless, the DNS-based postoperative implant position evaluation method is currently limited to the system used in the present study, and there is an urgent need to extend its application to other types of dynamic navigation systems. Additionally, the high equipment expenses, prolonged time of preoperative preparation, and the unavoidable learning curve of the DNS cannot be neglected [29]. Moreover, the present study is an in vitro study with a small sample size, and the comparison of the accuracy of the DNS-based method applied to free-end situations and single-tooth gaps was conducted based on different sample sizes. The results should also be confirmed in a more extensive dataset, an optimization of sample size allocation, and studies conducted in vivo.

## 5. Conclusions

This article presents a facile and efficient method for identifying implant position after surgery. This DNS-assisted method generates the 3D position of the realized implant without producing radiation or incurring additional costs. Within the limitations of this in vitro study, the results of the accuracy evaluation revealed that the described method yielded clinically acceptable deviations in comparison with CBCT scans, irrespective of the position and distribution of the inserted implants. Therefore, the described non-radiological method is reliable and possesses the potential to replace CBCT for evaluating implant positions after surgery, particularly for surgery conducted under the guidance of dynamic navigation. Moreover, it can be used to analyze the deviations between the planned and realized implant positions. Further investigations are required to verify its accuracy in vivo.

## Figures and Tables

**Figure 1 jpm-13-00362-f001:**
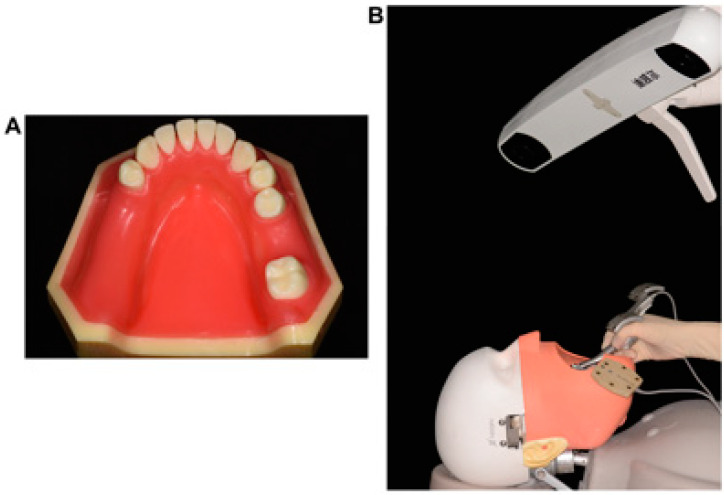
(**A**) Sample of the experimental model; (**B**) the plastic model was fixed onto a dental simulation unit before implant placement under the guidance of the dynamic navigation system.

**Figure 2 jpm-13-00362-f002:**
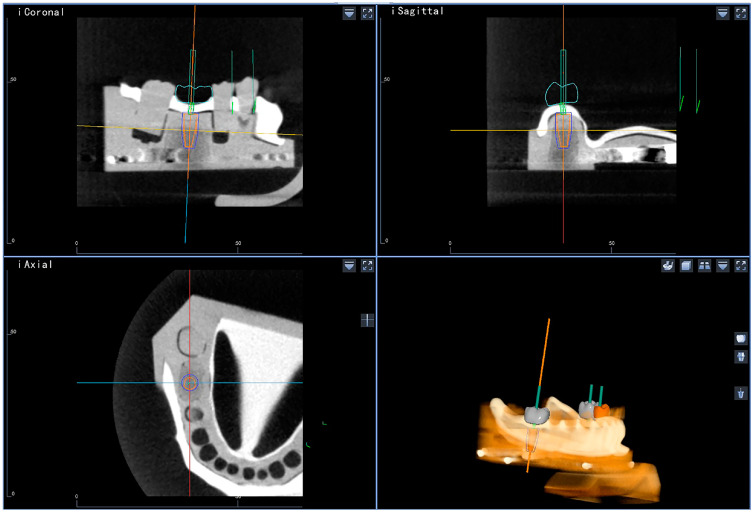
Preoperative treatment plan created by the dynamic navigation software.

**Figure 3 jpm-13-00362-f003:**
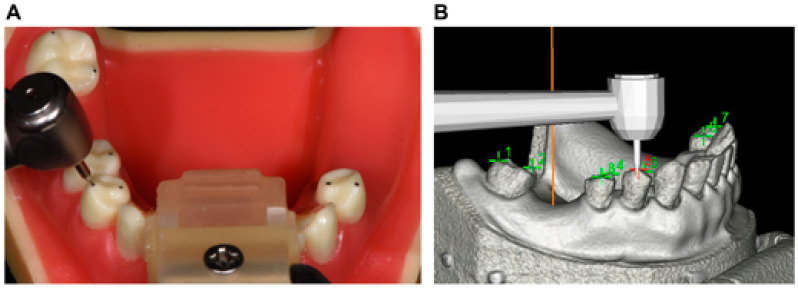
The process of cusp registration in oral (**A**) and software (**B**).

**Figure 4 jpm-13-00362-f004:**
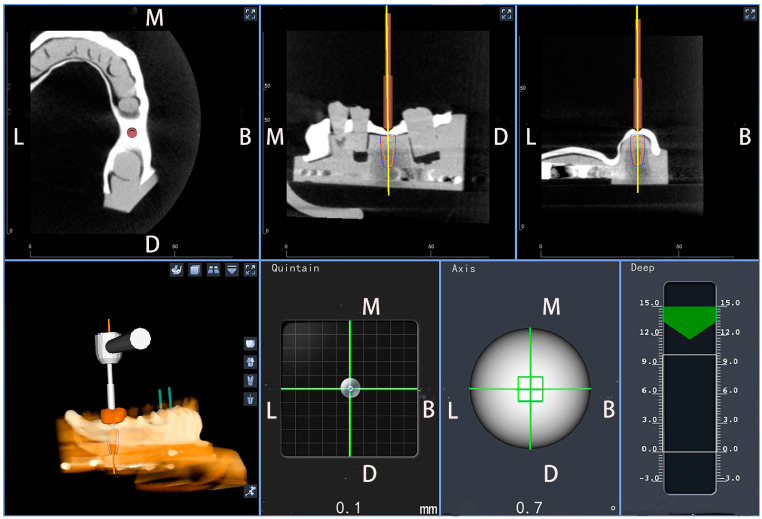
Work interface of the dynamic navigation software during surgical procedures.

**Figure 5 jpm-13-00362-f005:**
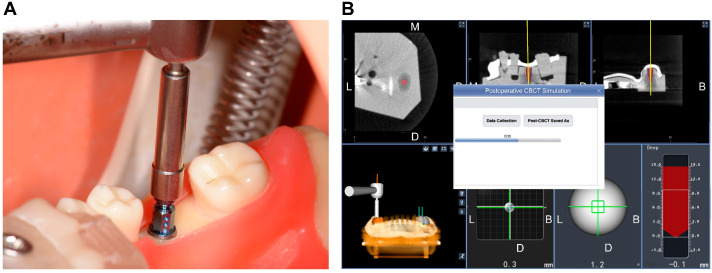
The process of collecting the three-dimensional positions of the inserted implants. The implant headpiece and implant carrier were used to collect the location information of the inserted implant (**A**); interface of the specially designed software that collected the information of the inserted implant (**B**).

**Figure 6 jpm-13-00362-f006:**
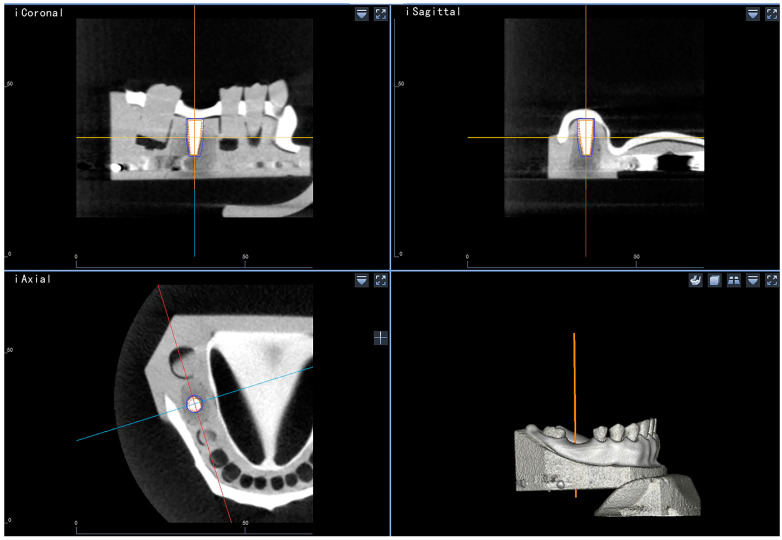
The achieved three-dimensional images of the virtual implant, which represented the actual implant in the dynamic navigation software.

**Figure 7 jpm-13-00362-f007:**
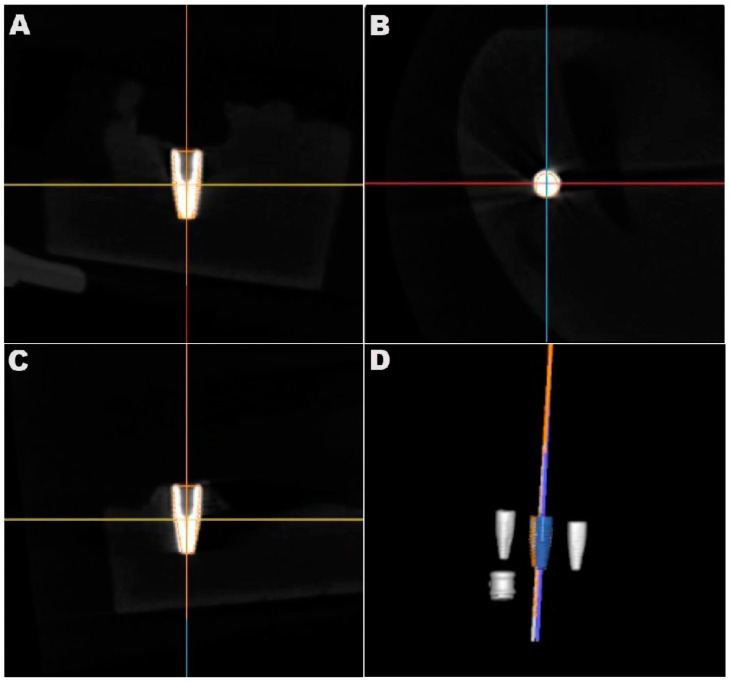
The reconstructed 3D images containing visible representations of the achieved implants generated by the DNS-based method (white implant in (**A**–**C**); blue implant in (**D**)) were superimposed on the postoperative CBCT dataset (orange line in (**A**–**C**); orange implant in (**D**)) for accuracy evaluation. (**A**) Implant sagittal; (**B**) implant coronal; (**C**) implant axial; (**D**) 3D image.

**Figure 8 jpm-13-00362-f008:**
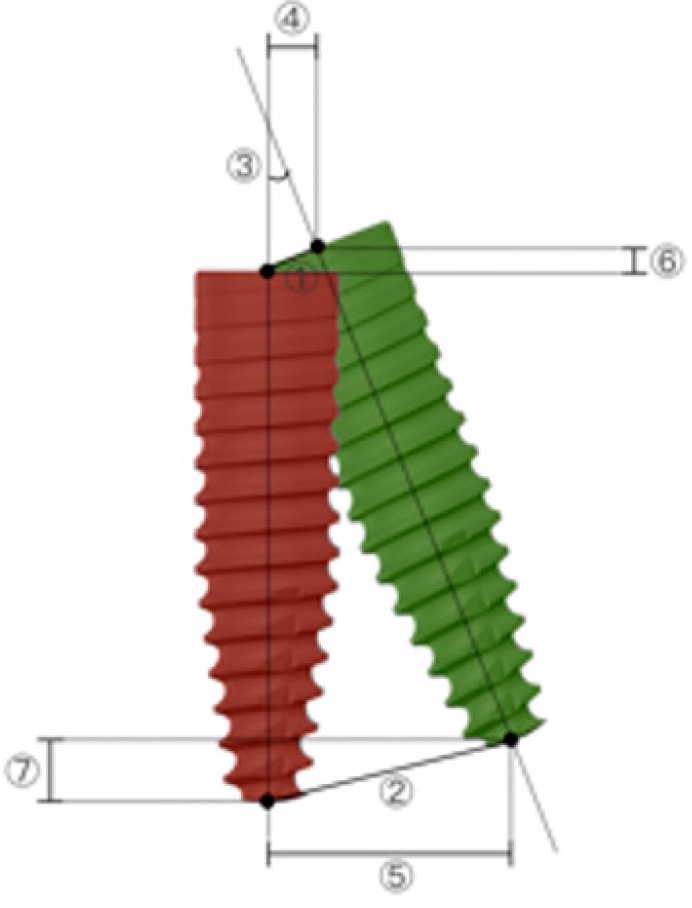
The target parameters measured in the deviations between the positions obtained by the DNS-based method (green color) and the postoperative CBCT (red color) results. ① Entry deviation (3D deviation at the implant shoulder); ② apex deviation (3D deviation at the implant apex); ③ angular deviation; ④ horizontal deviation at the entry point; ⑤ horizontal deviation at the apex point; ⑥ vertical deviation at the entry point; and ⑦ vertical deviation at the apex point.

**Table 1 jpm-13-00362-t001:** Measurements of deviation between the DNS-based evaluation method and postoperative cone beam computed tomography (CBCT).

	Minimum	Maximum	Mean	SD	95% CI	
					Lower	Upper
Entry deviation (mm)	0.32	1.66	0.88	0.37	0.78	0.97
Apex deviation (mm)	0.39	1.73	1.02	0.35	0.93	1.11
Angular deviation (°)	0.25	3.82	1.83	0.79	1.62	2.03
EH (mm)	0.07	1.22	0.59	0.26	0.53	0.66
AH (mm)	0.23	1.73	0.94	0.42	0.83	1.05
ED (mm)	−3.26	2.47	0.17	1.10	−0.11	0.45
AD (mm)	−3.23	3.43	0.32	1.31	−0.02	0.65

Abbreviations: SD—standard deviation; CI—confidence interval; EH—entry horizontal; AH—apex horizontal; ED—entry depth; AD—apex depth.

**Table 2 jpm-13-00362-t002:** Directions of horizontal deviation of the implant position obtained by the DNS-based method.

		BM	BD	LM	LD
Entry deviation	No.	19	9	15	17
	Value (mm)	0.59 ± 0.35	0.83 ± 0.48	0.6 ± 0.24	0.7 ± 0.39
Apex deviation	No.	11	10	20	19
	Value (mm)	0.94 ± 0.36	1.19 ± 0.6	0.97 ± 0.48	0.97 ± 0.34

Abbreviations: BM—buccal-mesial; BD—buccal-distal; LM—lingual-mesial; LD—lingual-distal.

**Table 3 jpm-13-00362-t003:** Difference of implant position deviations between implants placed in single-tooth gaps (STG) and free-end gaps (DE).

	Mann–Whitney U Test				
	Mean Difference (DE-STG)	SD	*p*-Value	95% CI	
				Lower	Upper
Entry deviation (mm)	0.07	0.10	0.433	−0.13	0.28
Apex deviation (mm)	0.04	0.09	0.520	−0.15	0.23
Angular deviation (°)	0.26	0.21	0.246	−0.17	0.69
EH (mm)	−0.04	0.07	0.655	−0.19	0.10
AH (mm)	0.03	0.12	0.724	−0.20	0.26
ED (mm)	0.46	0.30	0.224	−0.14	1.05
AD (mm)	0.70	0.35	0.122	0.01	1.40

Note: No significant difference was observed (*p* > 0.05).

**Table 4 jpm-13-00362-t004:** Difference of implant position deviations between different tooth positions (tooth no. 35 and tooth no. 36) at the free-end gap.

	*t*-Test				
	Mean Difference (35-36)	SD	*p*-Value	95% CI	
				Lower	Upper
Entry deviation (mm)	−0.06	0.12	0.624	−0.31	0.19
Apex deviation (mm)	−0.17	0.12	0.143	−0.41	0.06
Angular deviation (°)	−0.01	0.25	0.981	−0.52	0.51
EH (mm)	−0.02	0.08	0.778	−0.18	0.14
AH (mm)	0.09	0.13	0.494	−0.18	0.36
ED (mm)	−0.27	0.33	0.429	−0.94	0.41
AD (mm)	−0.70	0.39	0.089	−1.47	0.11

Note: No significant difference was observed (*p* > 0.05).

## Data Availability

Not applicable.

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
