# Peer review of "Proposal and Validation of a New Nonradiological Method for Postoperative Three-Dimensional Implant Position Analysis Based on the Dynamic Navigation System: An In Vitro Study"

_jpm, 2023, doi:10.3390/jpm13020362_

Round 1
Reviewer 1 Report
This study is aimed to assess the accuracy of a non-radiological method for postoperative 3D implant position analysis based on the Dynamic Navigation System (DNS). Overall is an interesting and well-written paper, but there are some minor issues to correct.
The title must include "in vitro" to be clear about the nature of the study.
The introduction must include a more detailed description of DNS, and its pros and cons.
Methods are well described, but a better graphical description of the implant placement under the DNS guide will be helpful to understand how the system assists the surgeon.
What are the results of your pre-experiment? Please be clear about how you estimate the sample size.
Regards the conclusion, please be more skeptical, this is preliminary in vitro study with a small sample.
Author Response
Point 1: The title must include “in vitro” to be clear about the nature of the study.
Response 1: Thank you very much for your considerable comment. We have added “in vitro” to our title.
Point 2: The introduction must include a more detailed description of DNS, and its pros and cons.
Response 2: Thanks a lot for your kind comment. We have added a detailed description of DNS and its pros (Introduction, Paragraph 3, Lines 13-17) and cons (Discussion, Page 14, Paragraph 1, Lines 1-2).
Point 3: Methods are well described, but a better graphical description of the implant placement under the DNS guide will be helpful to understand how the system assists the surgeon.
Response 3: Thank you very much for your kind comment. The clinical scenario and the software interface during the implant placement procedure, under the guidance of DNS, are shown in Figures 1B and 4, respectively. We believe these figures can help readers understand how DNS aids surgeons.
Point 4: What are the results of your pre-experiment? Please be clear about how you estimate the sample size.
Response 4: In our sample size calculation, the angular deviation was set as the primary outcome variable, the single-tooth gaps (STG) group as the control group, and the distal extension situations (DE) group as the test group. The results of our pre-experiment were as follows: the angular deviation of STG was 1.69 ± 0.59°, and that of DE was 2.21 ± 0.61°. Considering a power of 0.90, an alpha value of 0.05, a minimum sample size of 20 per group was calculated. According to the reviewer’s suggestions, we have added the results of the pre-experiment (Materials and Methods, Paragraph 2, Lines 3-4).
Point 5: Regards the conclusion, please be more skeptical, this is preliminary in vitro study with a small sample.
Response 5: Thank you very much for your valuable comment. According to your suggestion, we have refined the statements in our revised manuscript (Discussion, Page 14, Paragraph 1, Lines 3-4; Conclusions, Lines 3-4).
Reviewer 2 Report
Dear authors, I am delighted to do this review.
There are a few confounding parts in writing the paper.
The methodology must be improved.
It is not shown in the M and M section, the precise superimposition of the implant and virtual implant, so readers could not conclude how you achieved these differences. ( put proper images describing your precise measurements)
It is not stated within the text if the material used for guides is either biocompatible for humans or has the possibility for sterilization.
Thank you for your kind understanding
Author Response
Point 1: It is not shown in the M and M section, the precise superimposition of the implant and virtual implant, so readers could not conclude how you achieved these differences (put proper images describing your precise measurements).
Response 1: Thank you very much for your highly insightful suggestion. We have added a figure (Figure 7 in the revised manuscript) in the M and M section to show the precise superimposition of the implant position datasets obtained using the DNS-based method and postoperative CBCT.
Point 2: It is not stated within the text if the material used for guides is either biocompatible for humans or has the possibility for sterilization.
Response 2: Thanks a lot for your kind comment. The DNS-assisted implant position analysis was conducted without the need for guides. In this study, the materials in contact with the oral cavity included the 3D-printed splint and self-curing resin, which were used to fix the infrared tracking markers of the DNS onto the plastic models. The material of the 3D-printed splint is high-temp resin, which is the same as the material used for the static implant surgical guide. It is biocompatible with humans and can be sterilized. The self-curing resin is commonly used for fabricating the provisional crown. This, too, is biocompatible and is safe to use.
Reviewer 3 Report
The authors performed an in vitro study assessing the accuracy of navigation-based assessment for post-implantation positional assessment. 2 major concerns:
1. The authors aim was to assess the accuracy of the DNS system versus CBCT is assessing final implant position. The baseline for assessment was the pre-operative positional planning and the final benchmark of the position is the post-operative CBCT. As the authors pointed out, inaccuracy of the DNS guided implant insertion surgery might as well cause inaccurate insertion (compared to the planned positioned pre-operatively). So, the final implant position assesses by the DNS when compared to the final implant position in the CBCT is actually a sum of deviation from the 1. process of inserting of implant guided by DNS and 2. The process of measuring the implant position by DNS. I would have thought if the implant was inserted by guided splint (planned from pre-op CBCT) and assessed by DNS that would have eliminated this issue. Anyways, the authors must discuss this as a major limitation of the study.
2. The deviation of all parameters (except degree deviation) presented in table 1 is actually quite significant. All parameters maximum deviation is almost 2mm, some 3mm which in implant surgery perspective, very significant. So, the conclusion stating that “The results of the accuracy evaluation revealed that the described method yielded clinically acceptable deviations in comparison with CBCT scans” would not be accurate. The deviation of 2 to 3mm in implant surgery is not clinically acceptable deviation.
Author Response
Point 1:
The authors aim was to assess the accuracy of the DNS system versus CBCT in assessing final implant position. The baseline for assessment was the pre-operative positional planning and the final benchmark of the position is the CBCT. As the authors pointed out, inaccuracy of the DNS guided implant insertion surgery might as well cause inaccurate insertion (compared to the planned positioned pre-operatively). So, the final implant position assesses by the DNS when compared to the final implant position in the CBCT is actually a sum of deviation from the 1. process of inserting of implant guided by DNS and 2. The process of measuring the implant position by DNS. I would have thought if the implant was inserted by guided splint (planned from pre-op CBCT) and assessed by DNS that would have eliminated this issue. Anyways, the authors must discuss this as a major limitation of the study.
Response 1:
Thank you very much for your kind comment. In several previous studies[1,2] that used the intraoral scanning-assisted method to identify the positions of the placed implants, the accuracy was evaluated indirectly by calculating the differences between the data acquired by the intraoral scanning method (baseline: the preoperative positional planning; final benchmark: the final implant position assessed by the intraoral scanning method) and the postoperative CBCT (baseline: the preoperative positional planning; final benchmark: the position in the postoperative CBCT). These were indirect comparisons based on deviations between the planned and realized implant positions. In this indirect comparison method, the increased steps of alignment and potential errors derived from direction deviations may lead to decreased accuracy.
In the present study, however, the accuracy evaluation of the DNS-assisted implant position analysis method was conducted by measuring the deviations between the datasets obtained using the DNS-based method and postoperative CBCT directly. In this direct evaluation method, using the “gold standard” postoperative CBCT images as a reference to calculate the deviations of the DNS-assisted method may lead to a more accurate evaluation, with reduced superimposition times compared with the above-mentioned indirect method. Since this direct method did not involve comparing the preoperative positional planning and the postoperative position, the process of implant placement guided by the DNS did not contribute to the deviations.
However, the statement, “However, the deviation measured in DNS-guided implant surgery is a combined result of all errors in treatment planning and implant placement, including data acquisition, data processing, registration, and calibration. When DNS is used to identify the implant’s position after surgery, some other factors may also contribute to its inaccuracy, such as the algorithm of the specially designed software and connection stability between the implant carrier and the implant” in the original manuscript may lead to ambiguous interpretations. Therefore, we have refined the statements in the Discussion part of the revised manuscript (Page 13, Paragraph 1, Lines 6-13). Thank you again for pointing this out.
Point 2:
The deviation of all parameters (except degree deviation) presented in table 1 is actually quite significant. All parameters maximum deviation is almost 2mm, some 3mm which in implant surgery perspective, very significant. So, the conclusion stating that “The results of the accuracy evaluation revealed that the described method yielded clinically acceptable deviations in comparison with CBCT scans” would not be accurate. The deviation of 2 to 3mm in implant surgery is not clinically acceptable deviation.
Response 2:
Thank you very much for your highly insightful comment. As the “gold standard” for postoperative implant position analysis, CBCT may cause an average linear measurement error of 0.49 mm, due to the noise distortion and metallic artifacts around dental implants[3]. In the present study, the mean 3D deviation was 0.88 mm at the implant shoulder and 1.02 mm at the apex; the mean liner horizontal deviation was 0.59 mm at the implant shoulder and 0.94 mm at the apex; the mean liner vertical deviation was 0.17 mm at the implant shoulder and 0.32 mm at the apex. These mean deviations were almost in the same order of magnitude as that of CBCT. However, the range of deviations was relatively high, resulting in some high maximum deviations, especially for the liner vertical deviations (2.47 mm at the entry depth, and 3.43 mm at the apex depth). Nevertheless, their 95% confidence interval were -0.11–0.45 mm and -0.02–0.65 mm, respectively. This means that the vast majority of the deviations were kept in the same rank order as that of CBCT (0.49 mm). Based on this analysis, we conclude that, “The results of the accuracy evaluation revealed that the described method yielded clinically acceptable deviations in comparison with CBCT scans”.
However, unlike the liner vertical deviations, the standard deviations of the other 2D and 3D parameters were relatively normal. By inference, the high range of deviations of the liner vertical deviations might be attributed to the intrinsic errors of the DNS and specially designed software and remains to be improved with the optimization of the system and the newly designed software in our future work.
The reviewer’s comment is pertinent and rigorous, and we believe that with the improvement of the algorithm and upgrading of the system, the accuracy of the evaluation method will be improved further. Thank you again; we appreciated your valuable comment.
References
- Skjerven, H.; Olsen-Bergem, H.; Rønold, H.J.; Riis, U.H.; Ellingsen, J.E. Comparison of postoperative intraoral scan versus cone beam computerised tomography to measure accuracy of guided implant placement-A prospective clinical study. Clin. Oral Implants Res. 2019, 30, 531-541.
- Franchina, A.; Stefanelli, L.V.; Maltese, F.; Mandelaris, G.A.; Vantaggiato, A.; Pagliarulo, M.; Pranno, N.; Brauner, E.; Angelis, F.; Carlo, S.D. Validation of an Intra-Oral Scan Method Versus Cone Beam Computed Tomography Superimposition to Assess the Accuracy between Planned and Achieved Dental Implants: A Randomized In Vitro Study. Int. J. Environ. Res. Public Health 2020, 17, 9358.
- Al-Ekrish, AA.; Ekram, M. A comparative study of the accuracy and reliability of multidetector computed tomography and cone beam computed tomography in the assessment of dental implant site dimensions. Dentomaxillofacial. Radiol. 2011, 40, 67-75.
Round 2
Reviewer 3 Report
The authors responded to this reviewers comments and explained their findings well. I believe the current state of the manuscript following correction is ready for publication